**Data Availability Statement:** Regarding data exchange, the Ethics Committee of the La Princesa University Hospital approved this research without

# Use of hospital care services by chronic patients according to their characteristics and risk levels by adjusted morbidity groups

Jaime Barrio Cortes[1,2,3]*, María Martínez Cuevas[4], Almudena Castaño Reguillo[5], Mariana Bandeira de Oliveira[5], Miguel Martínez Martín[6], Carmen Suárez Fernández[6]

**1** Primary Care Investigation Unit, Madrid Health Service, Madrid, Spain, **2** Foundation for Biosanitary Research and Innovation in Primary Care, Madrid, Spain, **3** Faculty of Health, Camilo José Cela University, Madrid, Spain, **4** Fuencarral Healthcare Center, Madrid Health Service, Madrid, Spain, **5** Ciudad Jardín Healthcare Center, Madrid Health Service, Madrid, Spain, **6** Internal Medicine Department, La Princesa University Hospital, Madrid, Spain

* jaime.barrio@salud.madrid.org

## Abstract

### Background

In-hospital care of chronic patients is based on their characteristics and risk levels. Adjusted morbidity groups (AMG) is a population stratification tool which is currently being used in Primary Care but not in Hospitals. The objectives of this study were to describe the use of hospital services by chronic patients according to their risk levels assigned by AMG and to analyze influencing variables.

### Material and methods

In this cross-sectional study, patients aged ≥18 years from a healthcare service area classified as chronically ill by the AMG classification system who used their referral hospital services from June 2015 to June 2016 were included. Predisposing and needs factors were collected. Univariate, bivariate and multiple linear regressions were performed.

### Results

Of the 9,443 chronic patients identified (52.1% of the population in the selected area), 4,143 (43.9%) used hospital care services. Their mean age was 62.1 years (standard deviation (SD) = 18.4); 61.8% were female; 9% were high risk; 30% were medium risk, and 61% were low risk. The mean number of hospital service contacts was 5.0 (SD = 6.2), with 3.8 (SD = 4.3) visits to outpatient clinic, 0.7 (SD = 1.2) visits to emergency departments, 0.3 (SD = 2.8) visits to day hospital, and 0.2 (SD = 0.5) hospitalizations. The factors associated with greater service use were predisposing factors such as age (coefficient B (CB) = 0.03; 95% confidence interval (CI) = 0.01–0.05) and Spanish origin (CB = 3.9; 95% CI = 3.2–4.6). Among the needs factors were palliative care (CB = 4.8; 95% CI = 2.8–6.7), primary caregiver status (CB = 2.3; 95% CI = 0.7–3.9), a high risk level (CB = 2.9; 95% CI = 2.1–3.6), multimorbidity (CB = 0.8, 95% CI = 0.4–1.3), chronic obstructive pulmonary disease (COPD) (CB = 1.5,

considering the option of data sharing. Although the information is anomized and dissociated, there is clinical information about patients, so there are ethical and legal restrictions to sharing this data set. However, the data can be requested by contacting the Ethics Committee of the La Princesa University Hospital (ceim.hlpr@salud.madrid.org) or through the main researcher.

**Funding:** This project received a grant for the translation and publication of this paper from the Foundation for Biosanitary Research and Innovation in Primary Care (FIIBAP).

**Competing interests:** The authors have declared that no competing interests exist.

95% CI = 0.8–2.3), depression (CB = 0.8, 95% CI = 0.3–1.3), active cancer (CB = 4.4, 95% CI = 3.7–5.1), and polymedication (CB = 1.1, 95% CI = 0.5–1.7).

## Conclusions

The use of hospital services by chronic patients was high and increased with the risk level assigned by the AMG. The most frequent type of contact was outpatient consultation. Use was increased with predisposing factors such as age and geographic origin and by needs factors such as multimorbidity, risk level and severe diseases requiring follow-up, home care, and palliative care.

## Introduction

The progressively aging population [1, 2] and the increased prevalence of chronic diseases [3, 4] pose new challenges at the management level due to greater multimorbidity, functional impairment, and worse quality of life and lead to greater consumption of healthcare services, among other things [5, 6].

Spain's National Strategy for Addressing Chronicity [7] proposes using stratification models based on the Kaiser pyramid, which classifies chronic patients according to risk [8, 9]. At the bottom of the pyramid (low risk), where resource consumption and the need for services are lower, prevention and health promotion measures are focused on empowering patients and self-management. In the middle (medium risk), the need for care is greater, and management is determined according to the disease, with alternation between self-care and the use of health services, basically primary care services. At the top (high risk), where care consumption and needs are greatest, measures are directed toward case management relying on care by primary care and hospital care (S1 Fig) [10].

Different predictive models are available for resource consumption groups according to the complexity of their morbidities without incorporating other dimensions, such as socioeconomic status, disability, frailty, care, clinical parameters, or assessment scales [11]. Nevertheless, they have replaced systems based exclusively on demographic data. Currently, within these grouping models are Adjusted Clinical Groups (ACG) [12–15], Diagnostic Cost Groups (DCG-HCC) [12, 16], Community Assessment Risk (CRG) [12, 17], and Community Assessment Risk Screening (CARS) [18, 19]. These tools for grouping population morbidity have been integrated into the electronic health records (EHRs) of health systems to estimate the health resources consumed by each person and stratify patients into different levels of risk to determine the type of management and intervention required. In Spain, some previous tools have been applied and validated, new projects have been launched for multipathological patients (PROFUND/PALIAR) [20], and a morbidity grouper, the Adjusted Morbidity Groups (AMG), has been created in recent years from Spanish population data [21].

The AMG classifies the population into mutually exclusive groups based on morbidity and the level of complexity and has been introduced into the primary care EHRs of 13 autonomous communities as part of care strategies for patients with chronic diseases [21, 22].

The consumption of health services by the population is quantified according to the number of contacts with health personnel at the primary care, hospital care, and outpatient levels [23]. Chronic patients account for 80% of primary care consultations, 60% of admissions, 33% of emergency room visits and cost increases [19, 24, 25]. Several theoretical but not definitive models have been proposed, which attempt to explain the factors influencing the use of health

services and their relationships. Anderson established a theoretical basis from a behavioral model based on factors affecting the patient or user: predisposing factors independent of health status; needs factors dependent on health status; and factors that facilitate or hinder the need for health coverage after using health services [26].

Some studies have described the utilization of primary care services according to the AMG and associated factors [21, 27–29], but no study has specifically explored the utilization of hospital care services by patients stratified according to the AMG. This study could help to prove this grouper is adequate to be used in the Hospital for stratifying the chronic population at different risk levels to individualize their care and to improve the use efficiency of available resources. Therefore, the objective of this study was to describe the use of hospital services by chronic patients stratified according to the risk level by the AMG population morbidity grouper and to analyze predisposing factors and associated needs factors.

## Material and methods

### Design

This was a cross-sectional descriptive observational study with an analytical approach.

### Setting and study population

The population of this study included patients affiliated with the Ciudad Jardin healthcare center belonging to Primary Care Management of the Community of Madrid, resulting in a reference population consisting of 18,107 people and 9,443 chronic patients aged over 18 years. This health center is located in the district of Chamartín, Madrid, Spain, with 142,610 people enrolled in the municipal register, including 78,679 females (10,206 in Ciudad Jardin) and 8.8% foreigners (10.6% in Ciudad Jardin); their average age was 45.3 years (45.7 in Ciudad Jardin), and 21% were over 65 years old (22.9% in Ciudad Jardin) [30]. They had a deprivation index placing the territory in quartile 1, which pertains to neighborhoods with the lowest degree of deprivation in Madrid [31].

### Study subjects

Patients identified as chronic by the AMG stratification tool incorporated into the EHRs of the Community of Madrid who used the services of its referral hospital center (La Princesa University Hospital) between June 2015 and June 2016 were included. The Management Strategy for Chronic Patients of the Community of Madrid [7] considered any patient who presented at least one of the chronic diseases described in S1 Table as chronic. The 423 chronic patients under 18 years of age were excluded since La Princesa University Hospital primarily cares for adult patients. The AMG assignment is based on results for morbidities and the level of complexity. The morbidity grouper classifies the population into mutually exclusive groups based on the diagnostic codes recorded in EHR for each patient by the health professionals responsible for their care, and complexity is calculated by analyzing different variables, such as the risk of mortality, admissions, primary care visits, and prescriptions, that are linked to the patient's diagnoses and assigns the patient a numerical value (complexity index). This index allows the population to be stratified into three risk levels (high, medium, and low risk) as well as a subset without relevant chronic pathology (with four cutoff points obtained from the 40th, 70th, 85th, and 95th percentiles of the entire population) [21, 28, 29].

## Variables

The dependent variable was the number of annual contacts per patient in hospital care. The type of contact was defined as an emergency room visit, an outpatient clinic visit, admission or a day hospital. The independent variables were organized following the Andersen Behavioral Model [26]. Predisposing factors were sex, age, country of origin (Spain, Europe, and rest of the world), and the type of user according to health insurance card (active user: employed and unemployed: retired and without resources). The needs factors were being immobilized at home, being institutionalized in a nursing home, having a primary home caregiver, receiving palliative care, the number and type of chronic diseases, multimorbidity ($\geq$ 2 chronic diseases), the complexity index according to the AMG, and polymedication (patients with a medication regimen including five or more prescribed drugs for their chronic conditions, $\geq$ 5 active substances as baseline treatment). No variable acting as a facilitating factor of the service provider or related to the organization could be extracted.

## Source of information

The variables analyzed were extracted from information recorded by health professionals in the EHRs of the Community of Madrid primary care system. Sociodemographic and clinical care variables were collected on June 30, 2015. The use of hospital care services studied corresponded to the period between June 30, 2015, and June 29, 2016.

## Statistical analysis

A descriptive analysis of each qualitative variable expressed as a raw frequency and percentage was performed, and quantitative variables with a normal distribution are expressed as the mean and standard deviation (SD). To test the null hypothesis for the comparison of qualitative variables, the chi-squared test was used. To compare quantitative variables, the Mann-Whitney U-test and Kruskal-Wallis test were used. The Bonferroni method was used for multiple comparisons. The relationship between the number of annual hospital care contacts and the factors of the Andersen model was analyzed by linear regression. A model was constructed for the predisposing factors, another was established for the needs factors, and a final model including the statistically significant factors ($p<0.05$) in the two previous models was ultimately developed. The models were selected from among all possible models according to their consistency with the theoretical model and the principle of parsimony, that is, between two possible similar models, the simplest model (with the fewest assumptions) was selected. Data analysis was performed with the statistical software IBM SPSS Statistics version 25.

## Ethical and legal aspects

The analysis was performed on anonymized data, safeguarding patient confidentiality and adhering to current law. The study was approved by the Drug Research Ethics Committee of La Princesa University Hospital and received a favorable report from the Local Commission on Research of the Primary Care Management of the Community of Madrid.

## Results

### Predisposing factors and needs factors

Of the 9,443 chronic patients at the health center who were $\geq$ 18 years of age, 4,143 (43.9%) were identified to have used hospital services in the study period. Their mean age was 62.1 years (SD = 18.4), 2,559 (61.8%) were female, and 3,879 (93.6%) were of Spanish origin.

Regarding their stratification, 375 (9%) were classified by the AMG as high risk, 1,242 (30%) as medium risk, and 2,526 (61%) as low risk. Compared with medium- and low-risk patients, high-risk patients had a higher mean age (78.3 (SD = 11.6), 72.4 (SD = 13.5), and 54.7 (SD = 17.3, respectively), more chronic diseases (6.7 [SD = 2.4], 4.3 [SD = 1.5], and 2.1 [SD = 1.2], respectively), more multimorbidity (99.2%, 97.0%, and 61.3%, respectively), a higher complexity index (12.6, 2.8, and 2.2 respectively), more polymedication (81.9%, 43.9%, and 8.9%, respectively), more immobilization (25.6%, 6.0%, and 0.9%, respectively), more institutionalization (8.3%, 2.2%, and 1.2%, respectively), and a greater need for a primary home caregiver (21.1%, 4.9%, and 0.5%, respectively) (Table 1).

The most frequent chronic diseases within the high-risk category in the population of the healthcare area were arterial hypertension (84.3%), dyslipidemia (68.0%), cancer (38.1%), dysrhythmia (44.3%), diabetes mellitus (41.6%), obesity (29.9%), heart failure (29.3%), anemia (26.4%), chronic obstructive pulmonary disease (COPD) (25.3%), osteoarthritis (25.6%), ischemic heart disease (24.5%), thyroid disorder (24.3%), depression (23.5%), chronic kidney failure (22.4%), osteoporosis (22.7%), stroke (21.3%), and anxiety (20%). Among the most prevalent at all risk levels (high/medium/low), the pathologies that predominated in females were heart failure (29.3%/5.4%/0.2%) and anemia (26.4%/9.3%/7.0%), while in males, they were the cardiovascular risk factors arterial hypertension (84.4%/66.6%/27.6%), diabetes mellitus (41.6%/23.8%/7.0), dyslipidemia (68.0%/61.0%/35.7%), and obesity (29.9%/25.9%/14.7%), followed by dysrhythmias (44.3%/17.0%/2.9%), and COPD (25.3%/10.0%/2.2%) (Table 2).

**Table 1. Predisposing factors and needs factors of the total chronic patient population stratified by risk level and sex.**

| Level of risk<br>n (%) | Total 4143 (100) | High risk 375 (9) | Medium risk 1242 (30) | Low risk 2526 (61) | P | Female 2559 (61.8) | Male 1584 (38.2) | P |
|---|---|---|---|---|---|---|---|---|
| **Predisposing factors** | | | | | | | | |
| Sex Female | 2559 (61.8) | 198 (52.8) | 791 (63.7) | 1570 (62.2) | <0.01 | 2559 (100) | 0 (0) | 0.03 |
| Male | 1584 (38.2) | 177 (47.2) | 451 (36.3) | 956 (37.8) | | 0 (0) | 1584 (100) | |
| Age* | 62.1 (18.4) | 78.3(11.6) | 72.4 (13.5) | 54.7 (17.3) | 0.00 | 62.6 (18.8) | 61.5 (17.7) | 0.03 |
| ≤65 years | 2209 (53.3) | 57 (2.6) | 329 (14.9) | 1823 (82.5) | <0.01 | 1329 (60.2) | 880 (39.8) | 0.02 |
| >65 years | 1934 (46.7) | 318 (16.4) | 913 (47.2) | 703 (36.3) | | 1230 (63.6) | 704 (36.4) | |
| Origin Spain | 3879 (93.6) | 340 (90.7) | 1174 (94.5) | 2365 (93.6) | 0.03 | 2410 (94.2) | 1469 (92.7) | 0.01 |
| Europe | 74 (1.8) | 6 (1.6) | 23 (1.9) | 45 (1.8) | | 50 (2.0) | 24 (1.5) | |
| Rest of world | 190 (4.6) | 29 (7.7) | 45 (3.6) | 116 (4.6) | | 99 (3.9) | 91 (5.7) | |
| Active | 2170 (52.4) | 52 (2.4) | 316 (14.6) | 1802 (83) | <0.01 | 1314 (60.6) | 865 (39.4) | <0.01 |
| Retired | 1938 (46.8) | 305 (15.7) | 918 (47.4) | 715 (36.9) | | 1235 (63.7) | 703 (36.3) | |
| Without resources | 35 (0.9) | 18 (4.8) | 8 (0,6) | 9 (0,3) | | 10 (0,4) | 25 (1,6) | |
| **Needs factors** | | | | | | | | |
| Immobilized | 194 (4.7) | 96 (25.6) | 75 (6.0) | 23 (0.9) | <0.01 | 138 (5.4) | 56 (3.5) | <0.01 |
| Institutionalized | 88 (2.1) | 31 (8.3) | 27 (2.2) | 30 (1.2) | <0.01 | 65 (2.5) | 23 (1.5) | <0.01 |
| Primary caregiver | 153 (3.7) | 79 (21.1) | 61 (4.9) | 13 (0.5) | <0.01 | 106 (4.1) | 47 (3.0) | <0.01 |
| Home support | 50 (1.2) | 22 (5.9) | 24 (1.9) | 4 (0.2) | <0.01 | 31 (1.2) | 19 (1.2) | <0.01 |
| Palliative care | 35 (0.8) | 25 (6.7) | 5 (0.4) | 5 (0.2) | <0.01 | 16 (0.6) | 19 (1.2) | <0.01 |
| Chronic diseases* | 3.2 (2.1) | 6.7 (2.4) | 4.3 (1.5) | 2.1 (1.2) | <0.01 | 3.3 (2.1) | 3.0 (2.0) | <0.01 |
| Multimorbidity | 3126 (75.5) | 372 (99.2) | 1205 (97) | 1549 (61.3) | <0.01 | 1956 (76.4) | 1170 (73.9) | 0.06 |
| Index of complexity* | 9.5 (8.7) | 30.6 (12.6) | 12.7 (2.8) | 4.9 (2.2) | <0.01 | 9.3 (8.0) | 9.9 (9.7) | 0.56 |
| Polymedicated | 1073 (25.9) | 307 (81.9) | 541 (43.9) | 225 (8.9) | <0.01 | 708 (27.7) | 365 (23.0) | <0.01 |

* Mean (Standard deviation).

**Table 2. Distribution of the most prevalent chronic diseases in the population in the basic health area overall and stratified by risk level and sex.**

| Level of risk n (%) | Total 4143 (100) | High risk 375 (9) | Medium risk 1242 (30) | Low risk 2526 (61) | P | Female 2559 (61.8) | Male 1584 (38.2) | P |
|---|---|---|---|---|---|---|---|---|
| **Blood and immune system** | | | | | | | | |
| Anemia | 390 (9.4) | 99 (26.4) | 115 (9.3) | 176 (7.0) | <0.01 | 289 (11.3) | 101 (6.4) | <0.01 |
| HIV | 40 (1.0) | 5 (1.3) | 7 (0.6) | 28 (1.1) | 0.21 | 4 (0.2) | 36 (2.3) | <0.01 |
| Vasculitis | 16 (0.4) | 5 (1.3) | 4 (0.3) | 7 (0.3) | 0.01 | 11 (0.4) | 5 (0.3) | 0.56 |
| **Digestive system** | | | | | | | | |
| Cirrhosis | 270 (6.5) | 44 (11.7) | 133 (10.7) | 93 (3.7) | <0.01 | 138 (5.4) | 132 (8.3) | <0.01 |
| Inflammatory bowel disease | 45 (1.1) | 4 (1.1) | 17 (1.4) | 24 (1.0) | 0.51 | 26 (1.0) | 19 (1.2) | 0.58 |
| Liver disease | 250 (6.0) | 36 (9.6) | 120 (9.7) | 94 (3.7) | <0.01 | 131 (5.1) | 119 (7.5) | <0.01 |
| Pancreatitis | 6 (0.1) | 2 (0.5) | 3 (0.2) | 1 (0.0) | 0.04 | 3 (0.1) | 3 (0.2) | 0.55 |
| Ulcer | 90 (2.2) | 19 (5.1) | 30 (2.4) | 41 (1.6) | <0.01 | 37 (1.4) | 53 (3.3) | <0.01 |
| **Eyes and attachments** | | | | | | | | |
| Glaucoma | 247 (6.0) | 39 (10.4) | 106 (8.5) | 102 (4.0) | <0.01 | 161 (6.3) | 86 (5.4) | 0.26 |
| **Circulatory system** | | | | | | | | |
| Aortic aneurysm | 37 (0.9) | 15 (4.0) | 17 (1.4) | 5 (0.2) | <0.01 | 7 (0.3) | 30 (1.9) | <0.01 |
| Ischemic heart disease | 260 (6.3) | 92 (24.5) | 124 (10.0) | 44 (1.7) | <0.01 | 93 (3.6) | 167 (10.5) | <0.01 |
| Dysrhythmias | 449 (10.8) | 166 (44.3) | 211 (17.0) | 72 (2.9) | <0.01 | 257 (10.0) | 192 (12.1) | 0.04 |
| Hypertension | 184 (44.4) | 316 (84.3) | 827 (66.6) | 697 (27.6) | <0.01 | 1.084 (42.4) | 756 (47.7) | <0.01 |
| Stroke | 175 (4.2) | 80 (21.3) | 69 (5.6) | 26 (1.0) | <0.01 | 96 (3.8) | 79 (5.0) | 0.06 |
| Heart failure | 183 (4.4) | 110 (29.3) | 67 (5.4) | 6 (0.2) | <0.01 | 117 (4.6) | 66 (4.2) | 0.54 |
| Valvulopathy | 139 (3.4) | 72 (19.2) | 49 (3.9) | 18 (0.7) | <0.01 | 88 (3.4) | 51 (3.2) | 0.70 |
| **Locomotor system** | | | | | | | | |
| Arthritis | 139 (3.4) | 22 (5.9) | 69 (5.6) | 48 (1.9) | <0.01 | 101 (3.9) | 38 (2.4) | <0.01 |
| Osteoarthritis | 638 (15.4) | 96 (25.6) | 325 (26.2) | 217 (8.6) | <0.01 | 476 (18.6) | 162 (10.2) | <0.01 |
| Osteoporosis | 624 (15.1) | 85 (22.7) | 294 (23.7) | 245 (9.7) | <0.01 | 594 (23.2) | 30 (1.9) | <0.01 |
| **Nervous system** | | | | | | | | |
| Dementia | 127 (3.1) | 47 (12.5) | 54 (4.3) | 26 (1.0) | <0.01 | 91 (3.6) | 36 (2.3) | 0.02 |
| Epilepsy | 95 (2.3) | 21 (5.6) | 24 (1.9) | 50 (2.0) | <0.01 | 45 (1.8) | 50 (3.2) | <0.01 |
| Parkinson | 59 (1.4) | 15 (4.0) | 32 (2.6) | 12 (0.5) | <0.01 | 32 (1.3) | 27 (1.7) | 0.23 |
| **Psychological and psychiatric problems** | | | | | | | | |
| Alcohol abuse | 202 (4.9) | 37 (9.9) | 77 (6.2) | 88 (3.5) | <0.01 | 50 (2.0) | 152 (9.6) | <0.01 |
| Substance abuse | 57 (1.4) | 6 (1.6) | 21 (1.7) | 30 (1.2) | 0.43 | 24 (0.9) | 33 (2.1) | <0.01 |
| Anxiety | 1019 (24.6) | 75 (20.0) | 289 (23.3) | 655 (25.9) | 0.02 | 729 (28.5) | 290 (18.3) | <0.01 |
| Depression | 626 (15.1) | 88 (23.5) | 252 (20.3) | 286 (11.3) | <0.01 | 478 (18.7) | 148 (9.3) | <0.01 |
| Bipolar disorder | 45 (1.1) | 3 (0.8) | 16 (1.3) | 26 (1.0) | 0.66 | 26 (1.0) | 19 (1.2) | 0.58 |
| Psychotic disorder | 58 (1.4) | 4 (1.1) | 20 (1.6) | 34 (1.3) | 0.67 | 32 (1.3) | 26 (1.6) | 0.30 |
| **Respiratory system** | | | | | | | | |
| Asthma | 386 (9.3) | 25 (6.7) | 114 (9.2) | 247 (9.8) | 0.15 | 280 (10.9) | 106 (6.7) | <0.01 |
| COPD | 275 (6.6) | 95 (25.3) | 124 (10.0) | 56 (2.2) | <0.01 | 114 (4.5) | 161 (10.2) | <0.01 |
| **Endocrine system** | | | | | | | | |
| Diabetes mellitus | 629 (15.2) | 156 (41.6) | 296 (23.8) | 177 (7.0) | <0.01 | 308 (12.0) | 321 (20.3) | <0.01 |
| Dyslipidemia | 1915 (46.2) | 255 (68.0) | 758 (61.0) | 902 (35.7) | <0.01 | 1134 (44.3) | 781 (49.3) | <0.01 |
| Obesity | 805 (19.4) | 112 (29.9) | 322 (25.9) | 371 (14.7) | <0.01 | 494 (19.3) | 311 (19.6) | 0.80 |
| Thyroid disorder | 801 (19.3) | 91 (24.3) | 278 (22.4) | 432 (17.1) | <0.01 | 665 (26.0) | 136 (8.6) | <0.01 |
| **Urinary system** | | | | | | | | |
| Kidney failure | 116 (2.8) | 84 (22.4) | 26 (2.1) | 6 (0.2) | <0.01 | 58 (2.3) | 58 (3.7) | <0.01 |
| **Cancers** | | | | | | | | |

(*Continued*)

**Table 2.** (Continued)

| Level of risk n (%) | Total 4143 (100) | High risk 375 (9) | Medium risk 1242 (30) | Low risk 2526 (61) | P | Female 2559 (61.8) | Male 1584 (38.2) | P |
|---|---|---|---|---|---|---|---|---|
| Active cancer | 340 (8.2) | 143 (38.1) | 137 (11.0) | 60 (2.4) | <0.01 | 156 (6.1) | 184 (11.6) | <0.01 |
| Breast cancer | 47 (1.1) | 23 (6.1) | 16 (1.3) | 8 (0.3) | <0.01 | 46 (1.8) | 1 (0.1) | <0.01 |
| Prostate cancer | 60 (1.4) | 22 (5.9) | 30 (2.4) | 8 (0.3) | <0.01 | 0 (0) | 60 (3.8) | <0.01 |
| Colorectal cancer | 37 (0.9) | 12 (3.2) | 20 (1.6) | 5 (0.2) | <0.01 | 16 (0.6) | 21 (1.3) | 0.02 |
| Lung cancer | 26 (0.6) | 18 (4.8) | 7 (0.6) | 1 (0.1) | <0.01 | 10 (0.4) | 16 (1.0) | 0.01 |
| Lymphoma | 31 (0.7) | 13 (3.5) | 11 (0.9) | 7 (0.3) | <0.01 | 20 (0.8) | 11 (0.7) | 0.75 |
| Bladder cancer | 28 (0.7) | 14 (3.7) | 11 (0.9) | 3 (0.1) | <0.01 | 3 (0.1) | 25 (1.6) | <0.01 |
| Leukemia | 23 (0.6) | 9 (2.4) | 14 (1.1) | 0 (0) | <0.01 | 8 (0.3) | 15 (0.9) | <0.01 |
| Skin cancer | 40 (1.0) | 11 (2.9) | 15 (1.2) | 14 (0.6) | <0.01 | 27 (1.1) | 13 (0.8) | 0.45 |

At the in-hospital level, the most prevalent acute comorbidities in the high-risk group were infection and associated complications: recurrent urinary tract infections (18.7%), pneumonia (8.8%), severe sepsis (5.1%), and respiratory failure (4.3%) (Table 3).

**Use of services.** Of the 4,143 patients who used hospital care, the mean annual contacts totaled 5.0, which was higher in high-risk chronic patients than in medium- and low-risk chronic patients (10.0, 6.0, and 3.7, respectively). The most frequent type of contact was mainly outpatient consultations (6.7, 4.8, and 2.9, respectively), followed by emergency visits (1.5, 0.9, and 0.6), day hospitals (1.4, 0.3, and 0.2), and finally, hospitalizations (0.6, 0.2, and 0.1). In patients older than 65 years, the same trend continued for all chronic patients, with mean values higher than the mean for the population under 65 years of age: the mean number of outpatient consultations was 4.6 (vs. 4.3), followed by emergencies (1.4 vs. 0.9), day hospital visits (2.5 vs. 0.3), and finally hospitalizations (0.6 vs. 0.2) (Table 4).

The factors associated with greater service use in the final model were predisposing factors such as age (coefficient B (CB) = 0.03; 95% confidence interval (CI) = 0.01–0.05) and Spanish

**Table 3. Acute comorbidity distribution in the total population and stratified by risk level and sex.**

| Level of risk n (%) | Total 4143 (100) | High risk 375 (9) | Medium risk 1242 (30) | Low risk 2526 (61) | P | Female 2559 (61.8) | Male 1584 (38.2) | P |
|---|---|---|---|---|---|---|---|---|
| Repeated urinary tract infections | 275 (6.6) | 70 (18.7) | 96 (7.7) | 109 (4.3) | <0.01 | 221 (8.6) | 54 (3.4) | <0.01 |
| Pneumonia | 59 (1.4) | 33 (8.8) | 18 (1.4) | 8 (0.3) | <0.01 | 33 (1.3) | 26 (1.6) | 0.35 |
| Gastrointestinal bleed | 30 (0.7) | 12 (3.2) | 13 (1.0) | 5 (0.2) | <0.01 | 13 (0.5) | 17 (1.1) | 0.04 |
| Paralysis | 21 (0.5) | 15 (4.0) | 6 (0.5) | 0 (0) | <0.01 | 16 (0.6) | 5 (0.3) | 0.17 |
| Respiratory insufficiency | 19 (0.5) | 16 (4.3) | 2 (0.2) | 1 (0.1) | <0.01 | 12 (0.5) | 7 (0.4) | 0.90 |
| Severe sepsis | 21 (0.5) | 19 (5.1) | 2 (0.2) | 0 (0) | <0.01 | 7 (0.3) | 14 (0.9) | <0.01 |
| Disability | 15 (0.4) | 1 (0.3) | 3 (0.2) | 11 (0.4) | 0.615 | 6 (0.2) | 9 (0.6) | 0.08 |
| Tuberculosis | 15 (0.4) | 11 (2.9) | 1 (0.1) | 3 (0.1) | <0.01 | 11 (0.4) | 4 (0.3) | 0.36 |
| Nervous system infection | 4 (0.1) | 3 (0.8) | 1 (0.1) | 0 (0) | <0.01 | 2 (0.1) | 2 (0.1) | 0.63 |
| Femur fracture | 12 (0.3) | 6 (1.6) | 6 (0.5) | 0 (0) | <0.01 | 12 (0.5) | 0 (0) | <0.01 |
| Pneumothorax | 5 (0.1) | 4 (1.1) | 1 (0.1) | 0 (0) | <0.01 | 1 (0.0) | 4 (0.3) | 0.05 |
| Intestinal obstruction | 4 (0.1) | 3 (0.8) | 1 (0.1) | 0 (0) | <0.01 | 4 (0.2) | 0 (0) | 0.12 |
| Peritonitis | 1 (0.1) | 1 (0.3) | 0 (0) | 0 (0) | <0.01 | 1 (0.1) | 0 (0) | 0.43 |
| Gangrene | 1 (0.1) | 1 (0.3) | 0 (0) | 0 (0) | <0.01 | 1 (0.1) | 0 (0) | 0.20 |
| Spinal cord injury | 1 (0.1) | 0 (0.0) | 1 (0.1) | 0 (0) | 0.311 | 1 (0) | 0 (0) | 0.43 |
| Encephalitis | 3 (0.1) | 2 (0.5) | 0 (0) | 1 (0.1) | <0.01 | 1 (0) | 2 (0.1) | 0.31 |

**Table 4. Use of hospital care services in the total population of chronic patients and stratified by risk level and sex.**

| Risk levels n (%) | Total 4143 (100) | High risk 375 (9) | Medium risk 1.242 (30) | Low risk 2.526 (61) | P | Female 2559 (61.8) | Male 1584 (38.2) | P | Age ≤65 2209 (53.3) | Age >65 1934 (46.7) | P |
|---|---|---|---|---|---|---|---|---|---|---|---|
| **Total annual contacts*** | | | | | | | | | | | |
| | 5.0 (6.2) | 10.0 (11.9%) | 6.0 (6.0) | 3.7 (4.4) | <0.01 | 4.80 (5.0) | 5.4 (7.8) | 0.37 | 4.3 (5.7) | 5.7 (6.8) | <0.01 |
| **Place of contact*** | | | | | | | | | | | |
| Outpatient consultations | 3.8 (4.3) | 6.7 (6.8) | 4.7 (4.8) | 2.9 (3.1) | <0.01 | 3.6 (3.9) | 4.0 (4.9) | 0.45 | 3.3 (3.9) | 4.3 (4.6) | <0.01 |
| Emergency | 0.7 (1.2) | 1.5 (1.9) | 0.9 (1.4) | 0.6 (0.9) | <0.01 | 0.7 (1.2) | 0.7 (1.2) | 0.13 | 0.6 (1.1) | 0.9 (1.4) | <0.01 |
| Hospitalization | 0.2 (0.5) | 0.6 (0.9) | 0.2 (0.5) | 0.1 (0.3) | <0.01 | 0.1 (0.4) | 0.2 (0.5) | 0.34 | 0.1 (0.3) | 0.2 (0.6) | <0.01 |
| Day hospital | 0.3 (0.0) | 1.4 (5.7) | 0.3 (1.5) | 0.2 (2.6) | <0.01 | 0.2 (1.5) | 0.5 (4.1) | 0.88 | 0.3 (3.1) | 0.3 (2.5) | 0.6 |

* Mean (Standard deviation).

origin (CB = 3.9; 95% CI = 3.2–4.6). Among the needs factors were palliative care (CB = 4.8; 95% CI = 2.8–6.7), primary caregiver (CB = 2.3; 95% CI = 0.7–3.9), a high risk level (CB = 2.9; 95% CI = 2.1–3.6), multimorbidity (CB = 0.8, 95% CI = 0.4–1.3), COPD (CB = 1.5, 95% CI = 0.8–2.3), depression (CB = 0.8, 95% CI = 0.3–1.3), active cancer (CB = 4.4, 95% CI = 3.7–5.1), and polymedication (CB = 1.1, 95% CI = 0.5–1.7). The factors associated with lower utilization were being an active user of a health insurance plan (CB = -1.8; 95% CI = - 2.5–1.2) and being immobilized (CB = -3.4; 95% CI = - 4.8; -2) (Table 5).

## Discussion

Chronic patients have poorer functional status and greater morbidity and complexity, which lead to high consumption of hospital resources, which increases according to risk level [29, 32].

### Characteristics of the population

The studied population shared similar demographic characteristics and chronic disease prevalence rates oscillating between 40% and 60% in the province populations in Spain (approximately 40 and 50%) [29], America (38.9%) [33], Canada (47.8%) [34]), Australia (60%) [35], and Asia (43%) [36], as well as advanced age, female predominance, and the presence of polymedication, immobilization, and multimorbidity. The average age of the chronic patients in our series was higher than that of patients in another series described in a similar study (47.6) [37], probably due to a higher prevalence of immobility and because our population belonged to an area of the Autonomous Community of Madrid, where the average age tends to be higher than those in other regions (45.5 years). Female sex predominated, as in other series [36, 38], probably due to a longer life expectancy and thus a longer time to develop chronic diseases. However, as in other studies, the proportions of the two sexes were equal in the high-risk population, and this stratum of patients shares more chronic diseases, immobility, and polymedication and a greater probability of morbidity and mortality [39].

Of the 4,143 chronic patient users of hospital care services, 61% were low-risk patients, although this stratum has fewer needs for hospital care and should have an adequate longitudinal follow-up mainly by primary care doctors [40]. This high use of hospital services by patients at a lower risk suggests that such utilization may be related not only to the need for care but also the risk level. As their better functional state allows them to attend more follow-up or check-up visits on demand or referral appointments, often contradicting the follow-up strategy approach for chronic patients [7], repeat chronic pathology follow-ups through hospital visits should be managed exclusively by primary care. This study considered only patients

**Table 5.  Multivariate analysis of the variables correlated with the number of total contacts with HC.**

| Variables | Coef B | p | CI | |
|---|---|---|---|---|
| | | | Lower | Upper |
| **Predisposing factors model** | | | | |
| Sex | 0.63 | <0.01 | 0.25 | 1.02 |
| Age | 0.08 | <0.01 | 0.07 | 0.10 |
| Country of origin, Spain | 4.56 | <0.01 | 3.80 | 5.32 |
| Active user | -1.82 | <0.01 | -2.47 | -1.16 |
| **Needs factor model** | | | | |
| Immobilized | -3.35 | <0.01 | -4.79 | -1.91 |
| Palliative care | 5.25 | <0.01 | 3.25 | 7.25 |
| Primary caregiver | 2.26 | <0.01 | 0.67 | 3.86 |
| High risk level | 3.00 | <0.01 | 2.25 | 3.74 |
| Multimorbidity | 0.70 | <0.01 | 0.26 | 1.14 |
| Chronic obstructive pulmonary disease | 1.58 | <0.01 | 0.84 | 2.32 |
| Depression | 0.84 | <0.01 | 0.34 | 1.35 |
| Active cancer | 4.72 | <0.01 | 4.03 | 5.42 |
| Polymedicated | 0.72 | <0.01 | 0.26 | 1.19 |
| **Final model** | | | | |
| Age | 0.03 | <0.01 | 0.01 | 0.05 |
| Country of origin, Spain | 3.92 | <0.01 | 3.20 | 4.63 |
| Active user | -1.84 | <0.01 | -2.46 | -1.21 |
| Immobilized | -3.43 | <0.01 | -4.85 | -2.01 |
| Palliative care | 4.79 | <0.01 | 2.82 | 6.75 |
| Primary caregiver | 2.31 | <0.01 | 0.74 | 3.88 |
| High risk level | 2.86 | <0.01 | 2.12 | 3.59 |
| Multimorbidity | 0.84 | <0.01 | 0.37 | 1.30 |
| Chronic obstructive pulmonary disease | 1.55 | <0.01 | 0.82 | 2.30 |
| Depression | 0.71 | <0.01 | 0.21 | 1.20 |
| Active cancer | 4.41 | <0.01 | 3.73 | 5.10 |
| Polymedicated | 1.11 | <0.01 | 0.55 | 1.77 |

$R^2 = 0.16$.

who accessed hospital care without considering primary care visits. In addition, the AMG may not be as specific in predicting the use of hospital care compared to primary care visits since no studies have analyzed hospital visits with this grouper [27, 41].

The distribution according to the levels targeted in this study (high/medium/low) correlates with those described in the literature for conditions that are understood to be chronic. Patients described as high risk are characterized in the literature as frail, functionally impaired, and highly complex (1.4–5%) [36]; those at medium risk are described as having multimorbidity/ multiple pathologies (12.9–95.1%) [37]; and those at low risk are described as patients with a single chronic disease [29].

The prevalence of chronicity increases with age in both sexes, especially above age 65, with some differences depending on the region [42]. These differences are probably related to methodological limitations such as the absence of a single and homogeneous definition of chronic disease and the data included in the grouper, which can be overcome with more unified and common information [43]. This information would allow us to characterize the population worldwide and to perform better comparisons.

The stratification (high/medium/low risk) described by the AMG is a novel concept with high predictive value comparable to that of its alternatives (CRG, ACG) [22, 44]. Since the AMG uses only morbidity and complexity data, the addition of other concepts, such as the inclusion of socioeconomic status (deprivation index [31]), frailty, psychosocial profiles, and prognostic scales, would allow us to further improve the predictive value adjusted by the selected study population.

## Distribution of chronic diseases and acute hospital comorbidities

As in other studies in primary care [45, 46] and hospital care [21, 38, 47], the most frequent chronic diseases in patients were cardiovascular diseases and cancers, which were more serious diseases in patients at a higher risk. These diseases vary according to sex, with hypertension/chronic heart failure being more common in females and COPD, stroke, and ischemic heart disease being more common in males [45]. The most prominent acute in-hospital comorbidities were infections and their complications, similar to those described in the literature [48].

## Use of hospital care services

The average utilization rate of hospital care health services among the chronic patients was elevated at all risk levels; the rate was almost three times higher for high- vs. low-risk patients and two times higher for high- vs. medium-risk patients. Most contacts were outpatient consultations, followed by emergency department visits, day hospital visits, and finally, hospitalizations. No studies have compared patients stratified by the AMG by measuring only their use of hospital services since the AMG is predominantly integrated in primary care EHRs.

The use of health services depends mainly on user-related factors. Following the Andersen model, the factors that most determined the use of services in the final model were the predisposing factors of older age, as observed in other studies with AMG in adults in primary care [49], and Spain as the country of origin, which could be related to easier access to healthcare, although Spain has a universal public healthcare system that covers most of its population. Additionally, the needs factors of receiving palliative care and having a primary caregiver; presenting a high level of risk and multimorbidity; having diseases such as cancer, COPD, or depression; and being polymedicated were associated with greater use of services, possibly because all these factors are related to severe diseases requiring follow-up and extensive care. In contrast, being immobilized was correlated with lower health service use because these patients predominantly use primary care services, including nonretired patients [49]. It is estimated that 80% of the interventions performed in hospitals are directly related to chronicity, which also generates 77% percent of health expenditure. This is not an isolated event, it is a generalized problem and a rising challenge because two out of three deaths are directly related to these types of conditions [50].

Other studies in Europe, United States, Canada and Australia show that the use of health services and costs depends mainly on user-related factors [24, 51–54].

We did not address facilitating factors related to the health service organization due to a lack of data. However, the literature also attributes most service use to factors related to the user [55].

## Strengths and limitations

Among the limitations of the study are the methods and design selected. As a cross-sectional study, cause–effect relationships could not be established. The diagnostic coding and clinical assessment by the different professionals were not completely uniform and therefore limited our ability to objectively analyze the data. On the other hand, a proportion of patients may not

have been represented in the total population of the center since they have never contacted hospital care or visited the Private Health Services; thus, their profiles are not clearly reflected in the EHRs. However, considering that 95% of the population of the Autonomous Community of Madrid [56] has health insurance coverage, the remaining 5% is unlikely to significantly alter the results. Additionally, some predisposing factors (income level, education level, etc.) have been related to the use of health services in other studies [57], which we could not study here because information was not available. Despite all of the above limitations, through the use of EHR data, our study analyzes a large volume of real-world data on a population throughout treatment under real clinical practice conditions, thus minimizing the selection and memory biases typical of surveys.

On the other hand, regarding the use of services, the reasons for hospital care consultations could not be analyzed, nor did we determine whether these reasons were adequate or whether the patients should have initially visited a primary care provider. Some patients seek hospital care on their own or due to pressure from family members (emergencies or consultations) to schedule appointments sooner or for diagnostic testing (emergencies, day hospital, or consultations) given their availability and speed of results. They may also seek treatments such as transfusions, dialysis, plasmapheresis, and bleeding control (emergency/day hospital/outpatient visits) or need after-hours visits (because the healthcare center is closed). Another limitation related to the use of services was that we did not consider hospital consultations other than in-person consultations; thus, e-consultations, phone consultations, or home hospitalization as uses of hospital resources were not considered because they are not properly registered, even though such modalities are increasingly used in recent times. During the study period, these modalities were probably scarcer, but their use is currently increasing due to the COVID-19 pandemic, and they are becoming an essential means of consultation in times of confinement and isolation and can be an alternative to reduce overload on hospital care services.

Regarding the application of the AMG, authors have noted its limitation of focusing on clinical care management and not considering problems such as the psychosocial situation (disability, frailty, care) or economic status of patients (deprivation indexes) [31]; however, this limitation is shared by the other morbidity groupers. Nevertheless, Estupiñán et al. [22] demonstrated a strong predictive capacity of the AMG greater compared to that of other stratification tools. Thus, the AMG has been demonstrated to be useful for the stratification of chronic patients and to predict primary care service utilization [21, 41]. Therefore the Ministry of Health [7] in Spain aimed to apply them at a national scale to adapt to chronic patient treatment and related interventions (S1 Fig).

AMG and its implementation by the Ministry of Health in the different regions of Spain, has recently been selected by the WHO European Region as an example of good practice in the management of chronic patients by the health system so it could also be used in other European regions [29].

This study also shows that the AMG risk level can be useful to estimate hospital care service utilization, so hospital health professionals could benefit with this stratification of the chronic population at different risk levels based on AMG to offer a more individualized, coordinated and specific care and to improve the use efficiency of available resources. The utilization of AMG in Hospital settings would facilitate the coordination between the two main levels of care, a priority in chronic patients, as a means of achieving continuity of care, reducing costs and improving the quality of care. However, more studies are necessary to clarify this matter because this grouper has only been developed and studied in different regions in Primary Care settings in Spain [21, 27, 29].

In conclusion, the use of hospital services by chronic patients is high and increases with the risk level assigned by the AMG. The most frequent types of contact are outpatient

consultations and emergencies. Use was increased with predisposing factors such as age and geographic origin and needs factors such as multimorbidity, risk level, and severe diseases requiring follow-up, home care, and palliative care.

## Supporting information

**S1 Fig. Health professionals, tools and services for chronic patients management.** (JPG)

**S1 Table. Types of chronic diseases considered by the adjusted morbidity group (AMG) in the Community of Madrid at the time of data extraction.** (DOCX)

## Acknowledgments

We thank the professionals of the Ciudad Jardín healthcare center and the Research Unit of the Primary Care Management of Madrid for their methodological support.

## Author Contributions

**Conceptualization:** Jaime Barrio Cortes, María Martínez Cuevas, Mariana Bandeira de Oliveira, Miguel Martínez Martín, Carmen Suárez Fernández.

**Data curation:** Jaime Barrio Cortes, Mariana Bandeira de Oliveira.

**Formal analysis:** Jaime Barrio Cortes, Carmen Suárez Fernández.

**Funding acquisition:** Jaime Barrio Cortes.

**Investigation:** Jaime Barrio Cortes, Miguel Martínez Martín, Carmen Suárez Fernández.

**Methodology:** Jaime Barrio Cortes, María Martínez Cuevas, Almudena Castaño Reguillo, Mariana Bandeira de Oliveira, Carmen Suárez Fernández.

**Project administration:** Jaime Barrio Cortes, Carmen Suárez Fernández.

**Resources:** Jaime Barrio Cortes, Mariana Bandeira de Oliveira, Carmen Suárez Fernández.

**Software:** Jaime Barrio Cortes.

**Supervision:** Jaime Barrio Cortes, María Martínez Cuevas, Almudena Castaño Reguillo, Miguel Martínez Martín, Carmen Suárez Fernández.

**Validation:** Jaime Barrio Cortes, Almudena Castaño Reguillo, Mariana Bandeira de Oliveira, Carmen Suárez Fernández.

**Visualization:** Jaime Barrio Cortes, María Martínez Cuevas, Almudena Castaño Reguillo, Mariana Bandeira de Oliveira, Miguel Martínez Martín, Carmen Suárez Fernández.

**Writing – original draft:** Jaime Barrio Cortes, María Martínez Cuevas, Almudena Castaño Reguillo, Miguel Martínez Martín.

**Writing – review & editing:** Jaime Barrio Cortes, María Martínez Cuevas, Almudena Castaño Reguillo, Mariana Bandeira de Oliveira, Miguel Martínez Martín, Carmen Suárez Fernández.

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
