## [Decision Letter · Decision Letter 0]

27 Oct 2021

PONE-D-21-11752Use of hospital care services in chronic patients according to their characteristics and risk levels by Adjusted Morbidity Groups.PLOS ONE

Dear Dr. Barrio-Cortes,

Thank you for submitting your manuscript to PLOS ONE. After careful consideration, we feel that it has merit but does not fully meet PLOS ONE’s publication criteria as it currently stands. Therefore, we invite you to submit a revised version of the manuscript that addresses the points raised during the review process.

We look forward to receiving your revised manuscript.

Kind regards,

Sandra C. Buttigieg, MD PhD FFPH

Academic Editor

PLOS ONE

Journal Requirements:

Reviewers' comments:

Reviewer's Responses to Questions

**Comments to the Author**

1. Is the manuscript technically sound, and do the data support the conclusions?

Reviewer #1: Yes

Reviewer #2: Yes

2. Has the statistical analysis been performed appropriately and rigorously? 

Reviewer #1: Yes

Reviewer #2: Yes

3. Have the authors made all data underlying the findings in their manuscript fully available?

Reviewer #1: No

Reviewer #2: Yes

4. Is the manuscript presented in an intelligible fashion and written in standard English?

Reviewer #1: Yes

Reviewer #2: Yes

5. Review Comments to the Author

Reviewer #1: Thank you for this paper on predicating hospital care utilization in a region in Spain using a AMG grouper. Your paper is well structured and the analysis is presented in full detail. In your abstract I do miss the research gap that is being addressed by you research, why was it relevant to study the relationship between the AMG grouper and hospital care utilization? You do explain this to some degree in the discussion lines 315 through 318. In your introduction you may also consider to add more background information on how the results of your study will be used in addressing the problems that come with greater consumption of healthcare services (line 52-53). My other comments are:

1. I would recommend that you may include more or other literature, many of the studies that are now referred to were performed in Spain or written in Spanish. A suggestion would be to look at relevant studies conducted in the United States, Canada and Australia.

2. Also the role of not just hospital but also care utilization outside of the hospital by patients with chronic conditions should be discussed. You do mention that primary care utilization has been studied by other researchers. Perhaps you could make recommendations on what to do next now that the AMG grouper has been studied in relation to both primary and hospital care. Again the policy and/or clinical relevance of your findings are not directly clear from the paper.

3. I miss the reporting of missing data, was there missing data in your dataset. If yes, how was this handled?

4. To what degree are your results generalizable to other regions in Spain or other countries in Europe? From your discussion I understand the Ministry of Health in Spain may use your findings to apply national policies, however data are now derived from one single area in Madrid.

5. Did you consider not only calculating contacts per risk group but also the estimated costs of each contact? If it would be the aim of the Ministry of health to look at interventions to reduce or redistribute spending among risk groups? It would be helpful for the reader to have an better understanding what the ‘chronic patient treatment and related interventions’ mentioned in line 318 are precisely?

6. Furthermore please also consider to specify what ‘more studies’ in line 318 would be necessary to ‘clarify this matter’ in line 317-318, what do you mean exactly?

Reviewer #2: A very interesting paper and rigorously managed study. A very pertinent subject. Mostly well written. However, the writing style started to deteriorate along the way and needs to be thoroughly reviewed and edited especially from the Discussion onwards to improve its readability and flow.

6. PLOS authors have the option to publish the peer review history of their article (what does this mean?). If published, this will include your full peer review and any attached files.

Reviewer #1: No

Reviewer #2: No

---

## [Author Response · Author response to Decision Letter 0]

23 Nov 2021

Academic Editor:

Thank you very much for the Academic Editor comments.

We have made the minor essential revisions your reviewers suggested us and we have submitted the items required (Response to Reviewers, Revised Manuscript with Track Changes and Manuscript) into the PLOS ONE Editorial Manager Platform.

In case you need anything more please let us know.

Best regards,

Reviewer #1: 

- Thank you for this paper on predicating hospital care utilization in a region in Spain using a AMG grouper. Your paper is well structured and the analysis is presented in full detail. In your abstract I do miss the research gap that is being addressed by you research, why was it relevant to study the relationship between the AMG grouper and hospital care utilization? You do explain this to some degree in the discussion lines 315 through 318. 

We would like to thank the reviewer #1 for his/her comments. We are pleased that he/she thinks the paper is well structured and the analysis is presented in full detail.

We have included in the abstract more information to fill the research gap about why this study was relevant to study the relationship between the AMG grouper and hospital care utilization: “Adjusted morbidity groups (AMG) is a population stratification tool which is currently being used in Primary Care but not in Hospitals”.

- In your introduction you may also consider to add more background information on how the results of your study will be used in addressing the problems that come with greater consumption of healthcare services (line 52-53). 

Thank very much for the suggestion. We have added in our introduction more background information on how the results of our study will be used in addressing the problems that come with greater consumption of healthcare services: “This study could help to prove this grouper is adequate to be used in the Hospital for stratifying the chronic patients at different risk levels to individualize their care and to improve the use efficiency of available resources”.

- My other comments are:

1. I would recommend that you may include more or other literature, many of the studies that are now referred to were performed in Spain or written in Spanish. A suggestion would be to look at relevant studies conducted in the United States, Canada and Australia. 

Thank very much for the recommendation. Following the reviewer #1 suggestion we have erased studies performed in Spain or written in Spanish and we have added more references in order to show relevant studies written in English and performed in Europe[1–7], the United States[1,2,8–12], Canada[1,2,13,14] and Australia [1,2,15].

2. Also the role of not just hospital but also care utilization outside of the hospital by patients with chronic conditions should be discussed. You do mention that primary care utilization has been studied by other researchers. Perhaps you could make recommendations on what to do next now that the AMG grouper has been studied in relation to both primary and hospital care. Again the policy and/or clinical relevance of your findings are not directly clear from the paper.

We would like to thank the reviewer #1 for his/her comments. We have discussed the role of not just hospital but also care utilization outside of the hospital by patients with chronic condition taking in consideration primary care utilization that has been studied by other researchers. Besides, we have made some recommendations on what to do next now that the AMG grouper has been studied in relation to both primary and hospital care in the discussion section. We have tried to clarify policy and/or clinical relevance of our findings by stating in the discussion that “Hospital health professionals could benefit with this stratification of the chronic population at different risk levels based on AMG to offer a more individualized, coordinated and specific care and to improve the use efficiency of available resources. The utilization of AMG in Hospital settings would facilitate the coordination between the two main levels of care, a priority in chronic patients, as a means of achieving continuity of care, reducing costs and improving the quality of care”.

3. I miss the reporting of missing data, was there missing data in your dataset. If yes, how was this handled? 

No, there was no missing data in our dataset so there was no need to handle this matter.

4. To what degree are your results generalizable to other regions in Spain or other countries in Europe? From your discussion I understand the Ministry of Health in Spain may use your findings to apply national policies, however data are now derived from one single area in Madrid. 

The “Report of the Stratification of the Population by adjusted morbidity groups project in the National Health System” listed in the Introduction section of the manuscript [16]” provides some results regarding the use of AMG in Primary Care Settings, which proves our results could be generalizable to other regions in Spain because similar results in different regions of Spain are shown. We have included this reference again in the discussion section and we have stated: “this grouper has been developed and studied in different regions in Primary Care settings in Spain” [16].

In order to show it could be generalizable to other countries in Europe we have remarked in the discussion that: “AMG and its implementation by the Ministry of Health in the different regions of Spain, has recently been selected by the WHO European Region as an example of good practice in the management of chronic patients by the health system so it could also be used in other European regions as well[16].

We apologized if in the discussion it has been misunderstood that the Ministry of Health in Spain may use our findings to apply national policies. As the reviewer wisely stated, data are derived from one single area in Madrid so it´s not enough to apply national policies based only on this data. This is why we recommend that more studies focused in AMG in Hospital Care should be made to add more information regarding this matter, because this grouper has been only developed and studied in different regions in Primary Care settings in Spain.

5. Did you consider not only calculating contacts per risk group but also the estimated costs of each contact? If it would be the aim of the Ministry of health to look at interventions to reduce or redistribute spending among risk groups? It would be helpful for the reader to have an better understanding what the ‘chronic patient treatment and related interventions’ mentioned in line 318 are precisely?

No, we didn´t consider the estimated costs of each contact because we only had access to the number of contacts per risk group per year with no possibility to estimate the costs of these contacts.

However, to address this matter, we have added in the discussion that “It is estimated that 80% of the interventions performed in hospitals are directly related to chronicity, which also generates 77% percent of health expenditure. This is not an isolated event, it is a generalized problem and a rising challenge because two out of three deaths are directly related to these types of conditions” [17].

The Health Ministry take in account very seriously the number of interventions to reduce or redistribute spending among risk groups. The Project "Stratification of the Population of the SNS" is framed within the Strategy for Addressing Chronicity in the National Health Service, with the aim of providing a technological tool like AMG that allows the identification of population subgroups with different levels of need and risk, which can facilitate the provision of specific interventions to each need for chronic care, so that health care is adequate and efficient and the continuity of care is guaranteed. 

The “chronic patient treatment and related interventions” mentioned are specified with more detailed in the the strategies for addressing patients with chronic diseases in the different regions of Spain. In order to help the reader to have a better understanding, we have created a Supplementary S1 Figure in which appears all the health professionals, health tools and health services that the chronic patients ideally would need according to their risk levels. This is based on the Kaiser Pyramid care model and it is already explained in the introduction section: 

• Low-risk patients are patients with chronic conditions that are still in incipient stages. The goal for this level is to slow the progression of the disease and prevent the patient from reaching higher levels of risk. To this end, the self-management of the disease and the education of a preventive nature and healthy habits are supported and to avoid healthcare utilization.

• Medium-risk patients are patients who present chronic conditions that need a more disease-based approach. The goal, as at the intermediate level, is to slow progression by planning and managing the disease that combines self-management and professional care.

• High-risk patients are highly complex patients, with multimorbidity and a multidisciplinary care approach with an increase healthcare services utilization. The objective at this level is to reduce flare-ups and hospital admissions through comprehensive case management, with mainly professional care.

6. Furthermore please also consider to specify what ‘more studies’ in line 318 would be necessary to ‘clarify this matter’ in line 317-318, what do you mean exactly?

Thank you very much for this comment.

In that sentence we exactly mean that as this grouper has been only developed and studied in the Primary Care Settings, it is necessary more research like ours to have more evidence regarding the usefulness and applicability of AMG in Hospital Care.

We have clarified this sentence in the discussion adding the sentence: “because this grouper has been only developed and studied in the Primary Care Settings”.

Reviewer #2: 

- A very interesting paper and rigorously managed study. A very pertinent subject. Mostly well written. However, the writing style started to deteriorate along the way and needs to be thoroughly reviewed and edited especially from the Discussion onwards to improve its readability and flow.

We would like to thank the reviewer #2 for his/her comments. We are pleased that he/she thinks the paper is interesting and rigorously managed and with a very pertinent subject. 

As we are not native speakers, to improve the writing style, we have sent the paper again to America Journal Experts (AJE) to be thoroughly reviewed and edited especially from the Discussion onwards to improve its readability and flow. 

Also, we have tried to simplify some sentences to make it easier to understand and to facilitate its reading.

The changes we have made and the ones suggested by the editors from AJE are remarked in the tracked changes version along the manuscript.

We have attached the new AJE Editorial Certificate for English language, grammar, punctuation, spelling and overall style by highly qualified native English-speaking editors.

 

References

1. Marengoni A, Angleman S, Melis R, Mangialasche F, Karp A, Garmen A, et al. Aging with multimorbidity: A systematic review of the literature. Ageing Res Rev. 2011;10: 430–439. doi:10.1016/j.arr.2011.03.003

2. Huntley AL, Johnson R, Purdy S, Valderas JM, Salisbury C. Measures of Multimorbidity and Morbidity Burden for Use in Primary Care and Community Settings: A Systematic Review and Guide. Ann Fam Med. 2012;10: 134–141. doi:10.1370/afm.1363

3. Glynn LG, Valderas JM, Healy P, Burke E, Newell J, Gillespie P, et al. The prevalence of multimorbidity in primary care and its effect on health care utilization and cost. Fam Pract. 2011;28: 516–523. doi:10.1093/fampra/cmr013

4. Brilleman SL, Salisbury C. Comparing measures of multimorbidity to predict outcomes in primary care: A cross sectional study. Fam Pract. 2013;30. doi:10.1093/fampra/cms060

5. Sullivan CO, Omar RZ, Ambler G, Majeed A. Case-mix and variation in specialist referrals in general practice. Br J Gen Pract. 2005;55: 529–33. Available: http://www.ncbi.nlm.nih.gov/pubmed/16004738

6. Prinsze FJ, Van Vliet RCJA. Health-based risk adjustment: Improving the pharmacy-based cost group model by adding diagnostic cost groups. Inquiry. 2007;44: 469–480. doi:10.5034/inquiryjrnl_44.4.469

7. Barker I, Steventon A, Deeny SR. Association between continuity of care in general practice and hospital admissions for ambulatory care sensitive conditions: cross sectional study of routinely collected, person level data. BMJ. 2017;356: j84. doi:10.1136/bmj.j84

8. United States Census Bureau. Projections of the Population by Sex and Selected Age Groups for the United States: 2015 to 2060. [cited 2021 November 20]. Available: https://www.census.gov/data/tables/2014/demo/popproj/2014-summary-tables.html..

9. Adams EK, Bronstein JM, Raskind-Hood C. Adjusted clinical groups: predictive accuracy for Medicaid enrollees in three states. Health Care Financ Rev. 2002;24: 43–61. Available: http://www.ncbi.nlm.nih.gov/pubmed/12545598

10. Winkelman R, Mehmud S. A Comparative Analysis of Claims-Based Tools for Health Risk Assessment. Schaumburg; 2007. 

11. Dueñas-Espín I, Vela E, Pauws S, Bescos C, Cano I, Cleries M, et al. Proposals for enhanced health risk assessment and stratification in an integrated care scenario. BMJ Open. 2016;6: e010301. doi:10.1136/bmjopen-2015-010301

12. Shelton P, Sager MA, Schraeder C. The community assessment risk screen (CARS): identifying elderly persons at risk for hospitalization or emergency department visit. Am J Manag Care. 2000;6: 925–933. Available: http://www.ncbi.nlm.nih.gov/pubmed/11186504

13. Street M, Berry D, Considine J. Frequent use of emergency departments by older people: a comparative cohort study of characteristics and outcomes. Int J Qual Heal Care. 2018;30: 624–629. doi:10.1093/intqhc/mzy062

14. Nicholson K, Terry AL, Fortin M, Williamson T, Bauer M, Thind A. Prevalence, characteristics, and patterns of patients with multimorbidity in primary care: a retrospective cohort analysis in Canada. Br J Gen Pract. 2019;69: e647–e656. doi:10.3399/bjgp19X704657

15. Harrison C, Henderson J, Miller G, Britt H. The prevalence of diagnosed chronic conditions and multimorbidity in Australia: A method for estimating population prevalence from general practice patient encounter data. Ramagopalan S V., editor. PLoS One. 2017;12: e0172935. doi:10.1371/journal.pone.0172935

16. Ministerio de Sanidad, Servicios Sociales e Igualdad. Informe del proyecto de estratificación de la población por grupos de morbilidad ajustados (GMA) en el Sistema Nacional de Salud (2014-2016). [Internet]. 2018 [cited 2021 March 26]. Available from: https://www.mscbs.gob.es/biblioPublic/publicaciones/recursos_propios/resp/revista_cdrom/VOL94/ORIGINALES/RS94C_202007079.pdf

17. Bengoa R. Empantanados. Rev Innov Sanit y Atención Integr. 2008;1: 1–7. Available: http://pub.bsalut.net/risai/vol1/8

---

## [Decision Letter · Decision Letter 1]

3 Jan 2022

Use of hospital care services by chronic patients according to their characteristics and risk levels by adjusted morbidity groups.

PONE-D-21-11752R1

Dear Dr. Barrio-Cortes,

We’re pleased to inform you that your manuscript has been judged scientifically suitable for publication and will be formally accepted for publication once it meets all outstanding technical requirements.

Kind regards,

Sandra C. Buttigieg, MD PhD FFPH

Academic Editor

PLOS ONE

Additional Editor Comments (optional):

Reviewers' comments:

Reviewer's Responses to Questions

**Comments to the Author**

1. If the authors have adequately addressed your comments raised in a previous round of review and you feel that this manuscript is now acceptable for publication, you may indicate that here to bypass the “Comments to the Author” section, enter your conflict of interest statement in the “Confidential to Editor” section, and submit your "Accept" recommendation.

Reviewer #1: All comments have been addressed

Reviewer #2: (No Response)

2. Is the manuscript technically sound, and do the data support the conclusions?

Reviewer #1: Yes

Reviewer #2: (No Response)

3. Has the statistical analysis been performed appropriately and rigorously? 

Reviewer #1: Yes

Reviewer #2: (No Response)

4. Have the authors made all data underlying the findings in their manuscript fully available?

Reviewer #1: Yes

Reviewer #2: (No Response)

5. Is the manuscript presented in an intelligible fashion and written in standard English?

Reviewer #1: Yes

Reviewer #2: (No Response)

6. Review Comments to the Author

Reviewer #1: Thank you for adressing all of my remarks in your revised manuscript. Adding more information on the study's clinical and/orpolicy implications and adding figure 1 has clearified the paper. I think the paper can be accepted for publication in its' current form.

Reviewer #2: Thanks you for the revision. In my view you answered and/or addressed all recommendations submitted by the 2 reviewers

7. PLOS authors have the option to publish the peer review history of their article (what does this mean?). If published, this will include your full peer review and any attached files.

Reviewer #1: No

Reviewer #2: No

---

## [Editor Report · Acceptance letter]

25 Jan 2022

PONE-D-21-11752R1 

Use of hospital care services by chronic patients according to their characteristics and risk levels by adjusted morbidity groups. 

Dear Dr. Barrio Cortes:

I'm pleased to inform you that your manuscript has been deemed suitable for publication in PLOS ONE. Congratulations! Your manuscript is now with our production department. 

Kind regards, 

on behalf of

Professor Sandra C. Buttigieg 

Academic Editor

PLOS ONE